# Solar UV Radiation in the Tropics: Human Exposure at Reunion Island (21° S, 55° E) during Summer Outdoor Activities

**DOI:** 10.3390/ijerph17218105

**Published:** 2020-11-03

**Authors:** Jean-Maurice Cadet, Hassan Bencherif, Nicolas Cadet, Kévin Lamy, Thierry Portafaix, Matthias Belus, Colette Brogniez, Frédérique Auriol, Jean-Marc Metzger, Caradee Y. Wright

**Affiliations:** 1LACy, Laboratoire de l’Atmosphère et des Cyclones (UMR 8105 CNRS, Université de La Réunion, Météo-France), Saint-Denis de La Réunion 97744, France; hassan.bencherif@univ-reunion.fr (H.B.); kevin.lamy@univ-reunion.fr (K.L.); thierry.portafaix@univ-reunion.fr (T.P.); 2School of Chemistry and Physics, University of KwaZulu-Natal, Durban 4041, South Africa; 3Faculté de Lettres et Sciences Humaines, Université de La Réunion, Saint-Denis de La Réunion 97744, France; nicolas.p.cadet@gmail.com; 4Conseil Régional de La Réunion, 5 Avenue René Cassin, Sainte-Clotilde 97490, La Réunion, France; matthias.belus@cr-reunion.fr; 5Univ. Lille, CNRS, UMR 8518–LOA–Laboratoire d’Optique Atmosphérique, F-59000 Lille, France; colette.brogniez@univ-lille.fr (C.B.); frederique.auriol@univ-lille.fr (F.A.); 6Observatoire des Sciences de l’Univers de La Réunion, UMS 3365, 97744 Saint-Denis de La Réunion, France; jean-marc.metzger@univ-reunion.fr; 7Department of Geography, Geoinformatics and Meteorology, University of Pretoria, Pretoria 0002, South Africa; caradee.wright@mrc.ac.za; 8Environment and Health Research Unit, South African Medical Research Council, Pretoria 0001, South Africa

**Keywords:** solar UV radiation, tropics, UV exposure, human health, mountain, volcano, beach, hike

## Abstract

Reunion Island is a popular tourist destination with sandy beaches, an active volcano (Piton de la Fournaise), and Piton des Neiges, the highest and most dominant geological feature on the island. Reunion is known to have high levels of solar ultraviolet radiation (UVR) with an ultraviolet index (UVI) which can reach 8 in winter and 16 in summer (climatological conditions). UVR has been linked to skin cancer, melanoma, and eye disease such as cataracts. The World Health Organization (WHO) devised the UVI as a tool for expressing UVR intensity. Thresholds ranging from low (UVI 1–2) to extreme (UVI > 11) were defined depending on the risk to human health. The purpose of the study was to assess UVR exposure levels over three of the busiest tourist sites on the island. UVR was measured over several hours along popular hiking trails around Piton de la Fournaise (PDF), Piton des Neiges (PDN), and St-Leu Beach (LEU). The results were compared with those recorded by the local UV station at Saint-Denis. In addition, cumulative standard erythemal dose (SED) was calculated. Results showed that UVI exposure at PDF, PDN, and LEU were extreme (>11) and reached maximum UVI levels of 21.1, 22.5, and 14.5, respectively. Cumulative SEDs were multiple times higher than the thresholds established by the Fitzpatrick skin phototype classification. UVI measurements at the three study sites showed that Reunion Island is exposed to extreme UVR conditions. Public awareness campaigns are needed to inform the population of the health risks related to UVR exposure.

## 1. Introduction

The main natural source of ultraviolet radiation (UVR) on earth comes from the sun. UVR represents 5% of solar spectral irradiance. Despite this small percentage, UVR is both essential and dangerous for the biosphere. UVR wavelength range is 100–400 nm and is divided into three wavebands, depending on its transmission capability in the atmosphere and its biological effects on humans. UVA (315–400 nm) represents 95% of the UVR reaching Earth, most UVB (280–315 nm) is absorbed by the atmosphere, and UVC (100–280 nm) does not reach Earth’s surface due to the ozone layer [1,2]. When going through the atmosphere, UVR is subject to several amplitude variations. Surface UVR is modulated by several parameters: atmospheric parameters such as ozone, aerosols, or clouds [3,4,5]; geographic parameters (such as latitude or altitude); temporal parameters (seasons or time of the day, i.e., solar zenith angle [6]).

Surface UVR has beneficial effects on human health. Vitamin D synthesis requires UVR and is known to be an important factor to have healthy bones. Vitamin D also has a substantial impact on brain chemistry, for example, in brain serotonin levels which fight anxiety and depression [7]. In medicine, UVR has been used in phototherapy for decades [8,9]. Despite these positive effects, the adverse effects of UVR can be critical for human health. The characteristics of UVR increase the risk of sunburn, eye disease, such as cataracts, or immunodeficiency when people are overexposed to UVR. The skin carcinogenesis effect of UVR increases the risk of nonmelanoma skin cancer (NMSC) by DNA damage and rapid, abnormal increase of keratinocytes [10,11,12]. The risk related to UVR largely depends on skin phototype which characterizes sunburn susceptibility (Table 1 shows the Fitzpatrick skin phototype classification [13]) and also on protective measures such as adequate clothing, sunscreen, hats, or sunglasses [14,15].

Human behavior is a factor that can lead to overexposure to UVR. Indeed, the increase of outdoor activities synonymous with good health and sun tanning for aesthetic goals have increased the level of UVR exposure [16,17,18]. Artificial UVR sources also cause overexposure to UVR, such as welding torches and sunbeds. In 2015, the use of sunbeds was linked to 4% of melanoma in France [19]. Moreover, frequent use of sunbeds may become addictive, and subsequently increases UVR exposure where this behavior has been linked to acceleration of skin aging [20].

Reunion Island (Figure 1) is a tropical island in the southwest of the Indian Ocean, and is well known as a touristic destination due to its varied attractions such as subtropical rainforests, an active volcano, and lagoons. Listed as a world UNESCO heritage site since 2010, the wide biodiversity of endemic plants and animals generates a great interest for the destination [21]. In this context, more than 500,000 tourists (2018 data [22]) visit Reunion Island every year. Several outdoor activities are possible all year round due to the tropical climate. Tourists as well as the local population (+800,000) may experience intensive UVR exposure.

The World Health Organization (WHO) defined the UV index (UVI) as a simple tool for public awareness. The UVI starts from zero when there is no UVR and increases with UVR intensity. Different thresholds (i.e., 1–2: low, 3–5: moderate, 6–7: high, 8–10: very high, >11: extreme) have been defined depending on the risk to human health and the use of suitable protection [23]. Sea level climatological averages showed that the UVI can reach 8 during winter and 16 during summer on Reunion Island [24,25]. Reunion is a mountainous island where Cadet et al. [6] reported a maximum UVI of 20 at ~2900 m height. However, the UVI is not well known by the local population and there is even a low awareness of the extreme UVI dangers by the local dermatologists [26].

In Reunion Island, in 2015, the invasive melanoma rate was 6.1 cases per 100,000 people in a female standard population and 7.1 cases per 100,000 people in a male standard population, and a positive trend was found (statistics were not deemed comprehensive and no data have been available since 2015) [6,27].

In this context of high risk related to UVR, and taking into account the popularity of Reunion Island as a tourist destination, this study aims to assess the UVR exposure over three sites that are popular with local and foreign tourists throughout the year: the Piton de la Fournaise volcano (2630 m), the highest summit, Piton des Neiges (3070 m), and Saint-Leu Beach.

## 2. Materials and Methods

### 2.1. Instruments

#### 2.1.1. Solarmeter Model 6.5 UV Index Meter

A handheld Solarmeter Model 6.5 UV Index Meter (SN#10414) was used during the field experiments. This instrument is manufactured by Solarmeter^®^ (Glenside, PA, USA), a trademark of Solar Light Company Inc. A silicon carbide photodiode records erythemal UV irradiance in the 280 to 400 nm wavelength range. The instrument records erythemal-weighted UVR and the UVI as a proxy for exposure—that is more readily understood by the public—is calculated following the standard formula Equation (1) [28]. Erythemal-weighted UVR is obtained by integrated solar irradiance with erythemal action spectrum. The accuracy traceable to the National Institute of Standards and Technology (NIST) is 10% and a previous study [29] demonstrated a good correlation with the reference instrument. Before the start of the field UVI campaigns, the Solarmeter 6.5 instrument went through a side-by-side comparison test with the Bentham spectroradiometer in the SDN location, and appeared to overestimate the UVI values by 12%, which is coherent with the 10% and 5% accuracies given by the NIST for the Solarmeter and of the Bentham instruments, respectively.
(1)$$UVI=k_{er}.\int_{250\;nm}^{400\;nm}E_{\lambda }.S_{er}\left(\lambda \right).d\lambda ,$$
where k_er_ is a constant equal to 40 W^−1^.m^2^, E_λ_ is the solar spectral irradiance expressed in W.m^−2^.nm^−1^ for each wavelength (λ) measurement, S_er_ is the erythema action spectrum depending on λ, and dλ is the wavelength interval.

#### 2.1.2. Bentham Spectroradiometer DTMc300

UVI from a spectroradiometer Bentham DTMc300 was also used. This spectroradiometer is manufactured by Bentham Instrument Ltd. Co. (Reading, England, UK) and is operated by Observatoire de Physique de l’Atmosphère de la Réunion (OPAR) in Saint-Denis, Reunion. The instrument has been affiliated with the Network for the Detection of Atmospheric Composition Change (NDACC) since 2015. A calibration is performed every three months with a 150 W lamp and a 1000 W quartz tungsten halogen lamp from NIST. The wavelength misalignment correction is performed at a distance via self-made software [30] by the LOA (Laboratoire d’Optique Atmosphérique) from University of Lille-1, France. UVI is recorded every 15 min following the standard formula (Equation (1)) [28] using a four-minute wavelength scan in the 280–450 nm wavelength range. The instrument uncertainty is ±5% (coverage factor k = 2) [31].

### 2.2. Methods

Reunion Island is well known by tourists for its various landscapes and outdoor activities, from its sandy beaches to its mountainous inland areas where the highest summit of the island reaches 3070 m at the top of the Piton des Neiges. The numerous and well-maintained trails winding through the island make the island’s summits easily accessible for all nature enthusiasts all year round, making them among the busiest sites on the island. The Office National des Forêts (ONF—National Forestry Office) estimates that 1,200,000 hikers use the 850 km hike path network every year [32]. The Piton de la Fournaise, one of the world’s most active and accessible volcanoes, illustrates this passion for high-altitude hiking as it is the most visited place on the island, with approximately 400,000 visitors per year [32]. As for the beach area, it is located on the west coast of the island. This is because of the existence of a coral reef, which makes the shoreline attractive and very popular, especially on weekends and public holidays. The conditions to enjoy these sites are favorable all year as the island benefits from a tropical climate. It is in this context that we decided to assess the UVR exposure on some of the most visited sites to better comprehend what the visitors are exposed to along the numerous altitude hiking trails or when they spend time at the beach. We therefore chose to carry out our surveys on three popular sites: the hike that leads to one of the world’s most active volcanoes, Piton de la Fournaise (hereafter referred to as PDF), as the most visited natural site, the hike that allows visitors to climb up the Piton des Neiges (PDN), the highest summit of the island, and the beach in St-Leu (LEU), one of the most frequented beaches. These sites are shown in Figure 2.

The OPAR’s spectro-radiometer UVI data were used for this study. Located at Saint-Denis (hereafter referred to as SDN), the instrument affiliated to NDACC provides high-quality data, which have been used in several studies. We used UVI from this spectro-radiometer as a comparison to our field measurement campaign.

The field measurement campaign took place in December 2019. The measurement protocol used in this study was the same as in previous studies [6,33]. Ambient erythemal UVR measurements were made using a handheld Solarmeter Model 6.5 UV Index Meter (SN#10414). The latter does not allow measuring skin exposure for any part of the body. One measurement was made every 10 min by one of the volunteers following the supplier’s measurement recommendations. The operator held vertically the Solarmeter 6.5 sensor pointing to zenith while watching the display screen. Once stable, the UVI measurement was marked, in addition to an environmental indicator. However, error of vertical positioning of the instrument can induce measurement uncertainty. Cumulative standard erythemal doses (SEDs) were calculated from UVI following Equation (2). One SED is equivalent to 100 J.m^−2^.
(2)$$UV=\underset{i}{\sum }\frac{∆t\times UVI}{k_{er}\times 100},$$
where *UV_d_* is the cumulative dose (SED unit), *∆t* the interval time between two measurements, *k_er_* the constant defined in Equation (1), and 100 a factor for the conversion from [J.m^−2^] to [SED].

The UVI from the Bentham DTMc300 at the OPAR UV station (hereafter referred to as SDN) was compared to the experiments, only on an indicative basis as the different sites do not have the same environment and atmospheric conditions. Hourly mean and standard deviation Equations (3) and (4), respectively) of SDN UVI were computed using December 2019 data.
(3)$$\bar{UVI}=\;\frac{1}{n}\sum_{i=1}^{n}UVI_{i},$$
(4)$$\sigma =\sqrt{\frac{1}{n-1}\sum_{i=1}^{n}{\left(UVI_{i}-\bar{UVI}\right)}^{2}},$$
where $\bar{UVI}$ is the mean of UVI and *σ* the standard deviation, *n* is the number of UVI observations during the measurement campaign (December 2019).

## 3. Results and Discussion

The hike to PDF took place on 15 December 2019 and lasted 6 h (Figure 3a,b). The environment is completely exposed to the sun except during the first and the last 10 min, where short vegetation and shade can interfere with UVR measurement. The grey area at the bottom of Figure 3a highlights the presence of cloud (cumulus) from the typical diurnal cloud formation of the island [34,35]. Clouds may have reduced the UVR, but it could also be increased by cloud scattering [36]. The UVI was extreme (>11) for 3.5 h and reached a maximum value of 21.1. The total cumulative dose during this hike was 57 SED (Figure 3b) (Table 2).

The hike to PDN took place four days later on 19 December 2019 and lasted 8 h 30 (Figure 3c,d). Different environmental situations affected the UVR during this hike experiment. The first and the third grey areas (from 6:00 a.m. to 8:50 a.m. and from 1:00 p.m. to 2:30 p.m.) depicted at the bottom of Figure 3c show the presence of a dense tropical forest and shade where UVI was strongly attenuated. There was almost no direct sunlight during these periods of time. Other than these periods of respite, there was no vegetation and no shade which could affect UVR measurement. From 11:10 a.m. to 2:30 p.m., in the downward direction of the hike, UVI measurements were performed in the presence of cumulus cloud cover that stretched from the summit to the base of PDN. During the descent phase, UVI first increased due to increased cloud scattering, then decreased. The UVI was extreme (>11) for 3.5 h and reached a maximum value of 22.5. The total cumulative dose during this hike was 65 SED (Figure 3d) (Table 2).

The beach UVR campaign at the St-Leu site took place on 28 December 2019 and was the longest experiment in duration (i.e., 9 h). As seen from the grey shaded areas depicted at the bottom of Figure 3e, some morning clouds had developed and then quickly dissipated, thereby reducing UVR. The UVI was extreme (>11) during three consecutive hours and reached a maximum value of 14.5. The total cumulative dose during this experiment was 64 SED (Figure 3f) (Table 2).

Taking into account the differences in environmental and meteorological characteristics from one experimental site to another, in addition to differences in observation duration, these results cannot be intercompared. However, regardless of location and height, all experiments showed situations of very high and extreme exposure to UV radiation. As expected, the PDN hike showed the highest UV index (22.4) at the highest summit of the island (3070 m). Regarding exposure, the beach experience at St-Leu highlighted the longest exposure time. In fact, as seen in Figure 3e, we observed more than 4.5 h of continuous exposure to very high UVI values (higher than 8), during the local 10:00 a.m.–2:30 p.m. time period, resulting in a cumulative dose of ~50 SEDs. Moreover, for all sites, the obtained cumulative doses were extremely high (higher than 50 SED) (see Figure 3b,d,f).

UVR data at SDN showed a December 2019 UVR level higher than the December climatological UVR level (December averaged values derived over 11 years of continuous UVR observations from 2009 to 2019) (Figure 4). It was observed that the UVI values averaged over December 2019 were higher than the climatological values, but within one standard deviation. The orange shaded area in Figure 4a highlights a higher dispersion in the afternoon. As reported by Cadet et al. [24], the observed daily variability could be induced by cloud cover which was higher in the afternoon over the SDN site. This situation is explained by the position of Reunion Island in the east–west trade wind flow and the topography of the island [34,35,36].

Schoolchildren and outdoor workers are subject to UVR risk. An estimation of the possible exposure can be computed from the daily ambient UVR (Bentham DTMc300 spectro-radiometer) following Wright et al. [37]. Schoolchildren receive 5% of daily ambient UVR whereas outdoor workers receive 20%. Possible UVR exposures are 3 SED and 12 SED for schoolchildren and outdoor workers, respectively. These exposures can induce sunburn depending on skin phototype (Table 1). Several factors can influence the percentage of UVR. Previous studies showed that outdoor workers may experience 10% to 70% of the daily ambient UVR depending on their activity [38].

Météo-France provides UV index forecasting. The forecast is provided by the 3-D MOCAGE model [39]. The latter is a chemistry transport model which is able to reproduce multiple chemical, dynamical, and physical processes such as convection, chemistry, and climate interactions or emissions and depositions. It is used both for research and operational purposes at Météo-France. The UV index calculation in MOCAGE is based on lookup tables. These are precomputed tables generated with the TUV model for multiple values of ozone profile, altitude, solar zenith angle, surface albedo, and aerosols [40]. These tables are then used during the MOCAGE simulation in order to determine UV index at a model grid point for specific values of ozone, solar zenith angle, albedo, altitude, and aerosols [5]. Depending on the nebulosity, a correction is applied on the UV index value [41]. The maximum of UVI recordings made during the experiment at PDF and LEU were 21.1 and 14.5, while Météo-France UVI forecast was 18 and 16, respectively. This difference between the two sites is mainly due to the altitude. The maximum UVI recorded for the PDN experiment was 22.4, while UVI forecast at the beginning of the PDN hike (Cilaos village, 1200 m) was 16. On an indicative basis, by using UVI gradient with altitude [42], UVI forecast could be 20 at the PDN summit. Overall, during December 2019 at SDN, Météo-France forecasts underestimated UVI values by 10% in comparison to the OPAR Bentham DTMc300 spectro-radiometer. The UVI forecast bias with the handheld radiometer and the spectro-radiometer can be explained by (1) the instruments’ uncertainties, 10% and 5%, respectively, (2) the difficulties estimating the cloud effect on UVI (direct attenuation or enhancement by multiple scattering) in a tropical environment where cloud cover is highly variable, especially at altitude, (3) the difficulty for the model to take into account cloud covering at short space scales with high altitude variation, and (4) the difficulties of the MOCAGE model in estimating ozone levels in tropical regions of the southern hemisphere.

On an indicative basis, we used the UVI climatology at SDN (2009–2019) and altitude profile of the three experiments (PDF, PDN, and LEU) to compute the cumulative exposure doses. We used a UVI gradient from Blumthaler et al. (1997) [42]: +15.1%/1000 m at 60° solar elevation and + 18.6%/1000 m at 20° solar elevation. We obtained the cumulative exposure doses of 52, 67, and 51 SED for PDF, PDN, and LEU, respectively. These results were comparable to the cumulative exposure doses measured. However, these results should be taken with reservation as the use of a simple coefficient may not be sufficient [43]. UVI depends on other parameters such as ozone, cloud, albedo, and aerosols. The atmospheric chemical composition is very different on each side of the boundary layer [44,45,46]. The low aerosol load above the boundary layer induces higher UVR.

For comparison, we looked at the same time period of the three experiments, that is, 9 a.m. to 1 p.m. (6 h duration). The partial cumulative doses at PDF, PDN, and LEU were 53, 59, and 39, respectively. Differences between these results are compatible with the UVR gradient with altitude proposed by Blumthaler et al. (1997) [42].

## 4. Conclusions

Reunion Island is a popular tourist destination for its sandy beaches, active volcano, and mountains. Over a million people go through the hike path network every year. In this context where people are extremely exposed, UVR assessment was performed over three popular sites: the volcano Piton de la Fournaise, the highest summit Piton des Neiges, and Saint-Leu beach. These measurements took place during December 2019. Measurements showed that Reunion Island is exposed to extreme UVR conditions. The UVR assessment performed revealed that total UVR exposure can reach 65 SED during each individual popular activity. These exposure doses correspond to several times the minimal erythemal dose to elicit sunburn for each skin phototype. Public awareness of the risks related to UVR exposure becomes crucial when people do not use suitable protection since extreme UVR exposure can cause first-, second-, or third-degree burns and increase the risk of long-term physiological diseases [47]. Anecdotally, most people are usually not well informed of the dangers of long-term exposure to UVR especially in a tropical environment. Météo-France provides UVI forecasts [5], but this information is only available on the Météo-France website, unfortunately making them less accessible or known. This information would be more useful to the population if it was broadcast along with the daily weather (i.e., temperature, rainfall, wind, etc.) forecasts. As the entire population could be exposed to UVR when spending time outside, be it for professional or leisurely activities, it is fair to say that a daily UVI forecast would be in the public health interest. In the case of the UV experiment at Saint-Leu beach, for example, we observed about 4.5 h of continuous exposure to very high UVI values, between 10 h 00 min and 14 h 30 local time, resulting in a cumulative dose of ~50 SEDs. Therefore, the public must be informed of the danger of UV exposure and it should be recommended to avoid sunbathing and UV exposure during this time slot. Along with more awareness campaigns and more efficient measuring devices placed around the island, a general understanding of UVR will improve, thereby reducing the risks linked to excess exposure resulting from insufficient UVR protection (adequate clothing, sunglasses, hats, sunscreen). As the island’s trails and beaches are where tourists and locals are most exposed to UVR, signboards, placed in key areas, would be useful in raising awareness and improving people’s attitude towards UVR exposure risks. These notices, placed at the beginning of hiking trails and entrances to beaches, would summarize the risks and the protective measures that should be taken regarding solar UVR exposure. It is important to emphasize that Reunion observations and the above recommendations may be relevant to many other sites in tropical countries and territories.

## Figures and Tables

**Figure 1 ijerph-17-08105-f001:**
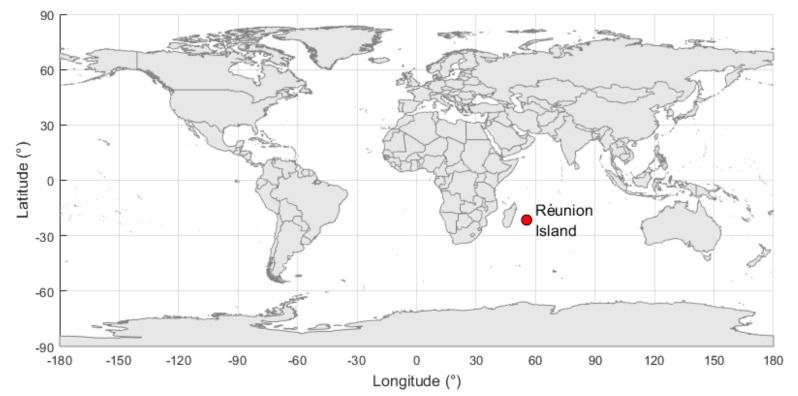
Reunion Island location.

**Figure 2 ijerph-17-08105-f002:**
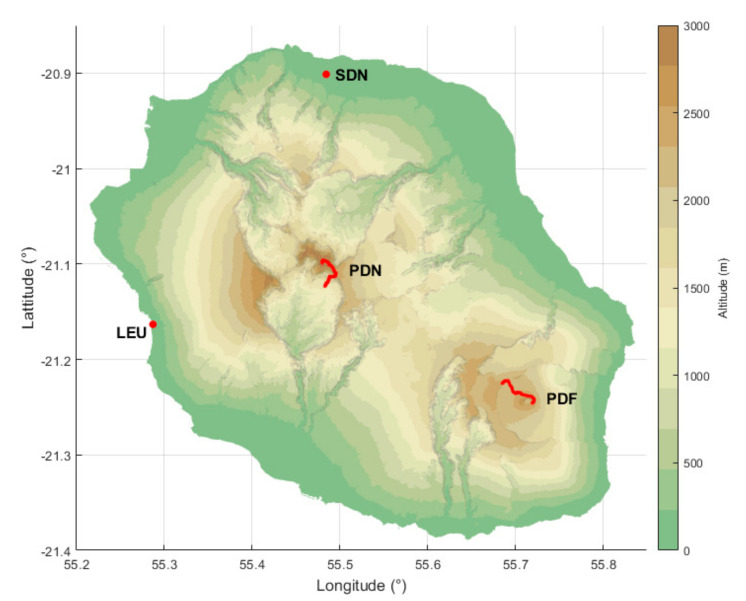
Measurement campaign locations at Reunion Island. The red dots show St-Leu Beach (LEU) and OPAR UV station at Saint-Denis (SDN). The red lines show the hike paths during the Piton de la Fournaise (PDF) and Piton des Neiges (PDN) hikes. The color bar on the right side displays the altitude in meters.

**Figure 3 ijerph-17-08105-f003:**
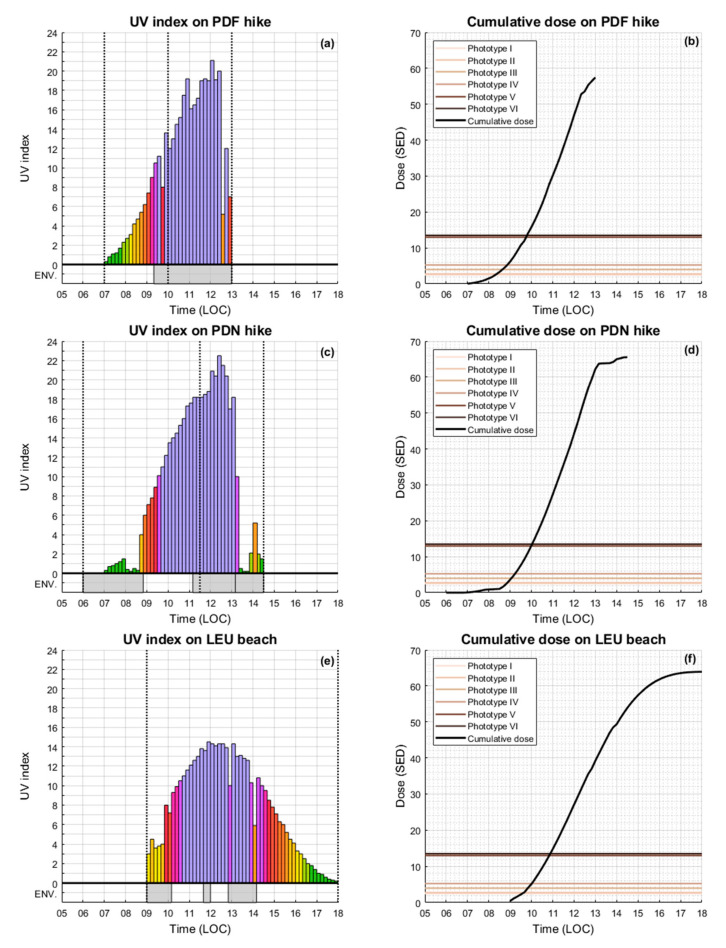
(**a**,**c**,**e**) show UVI recorded at PDF, PDN, and LEU, respectively. The colors of the histograms are similar to those used by the WHO, but with some variations within each color band to show more details. Vertical dotted lines show the beginning and the end of the activities. For PDF and PDN, a third vertical dotted line shows the U-turn point, between the ascending and descending part of the hikes. The grey surfaces, ENV., at the bottom of the figures show environmental effects affecting UVR, mainly cloud cover and shade. Cumulative UV doses at PDF, PDN, and LEU are shown in (**b**,**d**,**f**), respectively. The black curves on the right plots represent the cumulative exposure doses, while the superimposed horizontal lines show the threshold for one exposure dose to sunburn as a function of skin phototype (Table 1).

**Figure 4 ijerph-17-08105-f004:**
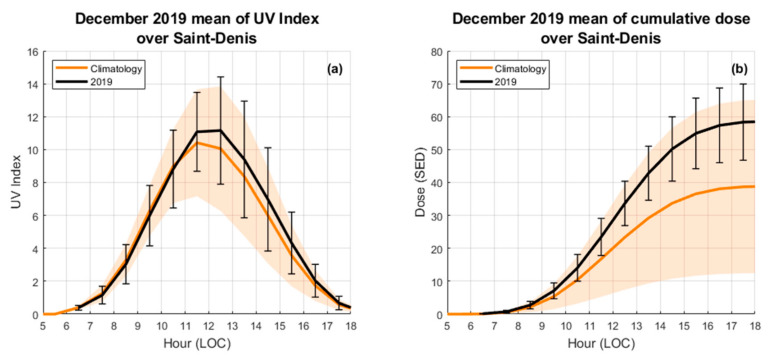
(**a**) represents a comparison of UVI between the December 2019 mean (black) and the December climatological mean (orange) at Saint-Denis; (**b**) represents a comparison of the cumulative exposure dose (SED) between the December 2019 mean (black) and the December climatological mean (orange) over Saint-Denis derived from UVI data (**a**). The orange lines represent the December climatological mean and the shaded areas one standard deviation as derived from 2009–2019 observations. The black lines represent the December 2019 mean and the vertical black bars one standard deviation.

**Table 1 ijerph-17-08105-t001:** Fitzpatrick skin phototype classification [13].

Phototype	Characteristics	History of Sunburn	Minimal Dose to Elicit Sunburn (SED)
I	Ivory white skin, light eyes	Burns easily	2–3
II	White skin, hazel/brown eyes	Burns easily	2.5–3
III	White skin, brown eyes	Burns moderately	3–5
IV	Light brown skin, dark eyes	Burns minimally	4.5–6
V	Brown skin, dark eyes	Rarely burns	6–20
VI	Dark brown skin, dark eyes	Never burns	6–20

**Table 2 ijerph-17-08105-t002:** Summary of the experiment results, with the associated cumulative doses.

Site	Type of Activity	Date	Duration	Altitude	Cumulative Dose
PDF	Hike	15 December 2019	6 h	2280–2480 m	57 SED
PDN	Hike	19 December 2019	8.5 h	1370–3070 m	65 SED
LEU	Beach	28 December 2019	9 h	1 m	64 SED
SDN	UV station	December 2019	11 h (full day)	85 m	59 SED

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
