# Peer review of "Solar UV Radiation in the Tropics: Human Exposure at Reunion Island (21° S, 55° E) during Summer Outdoor Activities"

_ijerph, 2020, doi:10.3390/ijerph17218105_

Round 1
Reviewer 1 Report
It is more or less a repetition of Ref [6] without any improvements or novelties (in Ref.6 it was a nice add-on).
The study design is not state of the art (it was already marginal in 2013 [Ref.:30]):
1) Experiments have to be repeated. It is very difficult if only one person does measurements with a hand-hold device. Was the person trained before and supervised afterwards?
2) The method of taking measurements is not described, just referenced. It must be described explicitly here. REF 6+30 are not very helpful: … done like recommended by the manufacturer…
This is essential to explain an UVI of 22.
Was only one single measurement done every 10 minutes?
It is very difficult to hold this meter vertically, pointing to zenith. Small inclinations directed to the sun give already noticeable higher values of irradiance than when directed to zenith.
3) The Solarmeter seems not to be proper calibrated. A description of the calibration procedure must be provided. This kind of meter is quite sensitive to spectral changes of solar radiation. If the meter is used as received from the manufacturer, uncertainties of up to 40% have to be considered. Easiest would have been to run the Solarmeter a few days in parallel with the Bentham. With that one could also distinguish between errors from spectral and angular response. To go from sea level up to 3000m will cause some additional uncertainties for both, because UV sky distribution as well as spectral distribution differs.
Quantities and units:
The integral of irradiance over time is radiant exposure, which is different from dose. Dose is radiant energy absorbed by mass or volume. In ionizing radiation radiant exposure and dose are quite similar, in the UV range there is a big difference (reflexion at the skin surface, absorption in the horny layer before striking the chromophore, backscattering,..., all these differ with body site).
Avoid mixing quantity and unit, for example
P1.,L.30: „In addition, cumulative Standard…. was calculated...“
SED is a unit.
Use instead: … cumulative radiant exposure expressed in SED was calculated.
p.1, L.31: SED is a unit (100J/m2) and is different from MED. Delete, „i.e.the minimal ….classification.“
See also:
- E. Braslavsky,
Glossary of terms used in photochemistry - 3rd Edition - IUPAC Recommendations 2006,
Pure Appl. Chem., 2007, 79, 293–465.
Or
David H. Sliney
Radiometric Quantities and Units Used in Photobiology and Photochemistry: Recommendations of the Commission Internationale de l’Eclairage (International Commission on Illumination)
Photochemistry and Photobiology, 2007, 83: 425–432
If you consider revision and resubmission:
The manuscript is well written. There is a lot of detailed information especially about the island and the environment which is very helpful to get an impression of the UV radiation environment there. The figures are very well done, are clear and easy to understand.
After problems (see above) have solved:
Chapter 3 Results and Discussion contains mainly discussion. A comparison between a UV Index forecast for a small island and hand-held measurements on the top of a mountain is rather tricky, especially when hand-held measurements are not really trustable.
A clear explanation must be given to explain 14UVI at the beach and 22UVI at 3000m.What have been the Bentham measurements on these days?
Possible additions:
Why not making hand-held measurements near the Bentham? This will give you clear estimate of variability due to hand-held. Be honest with uncertainties. Well-calibrated SolarLight Model 500 is within 10%, 15% when used operationally. UV Index of 22 will not become lower, because uncertainty goes in both directions, but more credible.
Why not using additionally UV-Index-hours? Especially a second scale/column to Fig.3 and Tab.2? Could give a clearer connection to the UV-Index (65 SED * 10/9 = 72 UVIh)
Saxebøl, UVIh—a proposal for a practical unit for biological effective dose for ultraviolet radiation exposure, Radiat. Prot. Dosim., 2000, 88, 261
Ratio of personal UV expose to ambient UV?
(School Children and outdoor workers (P.6, L.209) are maybe not the target groups for a 3000m hike)
What about sun protection? SPF for hats?
Diffey and Cheesman, BJD 127 (1) 1992 10-12
Author Response
1 October 2020
Open Review
Dear Reviewer,
First of all, I would like to thank you for your work on our paper. Our response is structured based on your review. We identified ten comments, referred hereafter as C1 to C10 and shown in black. Our responses are indeed labelled from R1 to R10 and printed in blue.
To make the reading of the revised manuscript easier, all edits are highlighted in the manuscript in yellow.
With kind regards,
Jean-Maurice CADET
It is more or less a repetition of Ref [6] without any improvements or novelties (in Ref.6 it was a nice add-on).
The study design is not state of the art (it was already marginal in 2013 [Ref.:30]):
C1: Experiments have to be repeated. It is very difficult if only one person does measurements with a hand-held device. Was the person trained before and supervised afterwards?
R1: We do agree; the experiments are repetitive. They must follow a specific protocol in order to ensure similar experimental conditions and time-sampling. Before each field campaign, the operator reads and discusses the experimental protocol and is trained and supervised on the use of the Solarmeter 6.5 instrument.
C2: The method of taking measurements is not described, just referenced. It must be described explicitly here. REF 6+30 are not very helpful: … done like recommended by the manufacturer…
R2: Description of the method of taking measurement is now improved in the revised version (see L156-159). The operator should hold vertically the Solarmeter 6.5 sensor pointing to zenith while watching the display screen. Once stable, the UVI measurement is marked, in addiction to an environmental indicator. However, error of vertical positioning of the instrument can induce measurement uncertainty.
C3: Was only one single measurement done every 10 minutes?
R3: One measurement was made every 10 minutes by the operator following the supplier measurement recommendations.
C4: It is very difficult to hold this meter vertically, pointing to zenith. Small inclinations directed to the sun give already noticeable higher values of irradiance than when directed to zenith.
R4: The verticality of the Solarmeter 6.5 instrument during the measurement is one of the inherent difficulties of the experimental protocol. It can in fact induce uncertainties in irradiance values. In order to reduce this uncertainties, the operator should hold vertically the Solarimeter 6.5 sensor pointing to zenith while watching the display screen, and the measurement is marked when the UVI value displayed is stable.
C5: The Solarmeter seems not to be proper calibrated. A description of the calibration procedure must be provided. This kind of meter is quite sensitive to spectral changes of solar radiation. If the meter is used as received from the manufacturer, uncertainties of up to 40% have to be considered. Easiest would have been to run the Solarmeter a few days in parallel with the Bentham. With that one could also distinguish between errors from spectral and angular response. To go from sea level up to 3000m will cause some additional uncertainties for both, because UV sky distribution as well as spectral distribution differs.
R5: Before the start of the field UVI campaigns, our Solarimeter instrument went through a side-by-side comparison test with the Bentham spectroradiometer in the SDN location. We did not report on this in the first version of the manuscript. This is now added in the revised version, and the calibration of the Solarmeter with the Bentham appeared to overestimate the UVI values by 12%, which is coherent with the 10% and 5% accuracies given by the NIST (National Institute of Standards and Technology) for the Solarmeter and of the Bentham instruments, respectively. (see L111-114)
C6: The integral of irradiance over time is radiant exposure, which is different from dose. Dose is radiant energy absorbed by mass or volume. In ionizing radiation radiant exposure and dose are quite similar, in the UV range there is a big difference (reflexion at the skin surface, absorption in the horny layer before striking the chromophore, backscattering,..., all these differ with body site).
Avoid mixing quantity and unit, for example
P1.,L.30: „In addition, cumulative Standard…. was calculated...“
SED is a unit.
Use instead: … cumulative radiant exposure expressed in SED was calculated.
p.1, L.31: SED is a unit (100J/m2) and is different from MED. Delete, „i.e.the minimal ….classification.“
R6: All of these recommended changes have been amended as requested.
C7: See also:
- Braslavsky, Glossary of terms used in photochemistry - 3rd Edition - IUPAC Recommendations 2006, Pure Appl. Chem., 2007, 79, 293–465.
Or
David H. Sliney, Radiometric Quantities and Units Used in Photobiology and Photochemistry: Recommendations of the Commission Internationale de l’Eclairage (International Commission on Illumination), Photochemistry and Photobiology, 2007, 83: 425–432
R7: Thank you for the references. We have added them to the manuscript.
If you consider revision and resubmission:
The manuscript is well written. There is a lot of detailed information especially about the island and the environment which is very helpful to get an impression of the UV radiation environment there. The figures are very well done, are clear and easy to understand.
After problems (see above) have solved:
C8: Chapter 3 Results and Discussion contains mainly discussion. A comparison between a UV Index forecast for a small island and hand-held measurements on the top of a mountain is rather tricky, especially when hand-held measurements are not really trustable.
R8: It is true that the Solarmeter 6.5 instrument does not offer the same measurement quality as the Bentham can do. The Solarmeter is a manual instrument designed to take measurements in a variety of environments. Its portability is both an advantage and a disadvantage.
In order to reduce uncertainties, the instrument was calibrated during a side-by-side comparison test with the Bentham (see R5 to C5). In addition to that, as mentioned in R1 to C1, in order to ensure similar experimental conditions and time-sampling, an experimental protocol was set up and followed by the operator.
C9: A clear explanation must be given to explain 14 UVI at the beach and 22 UVI at 3000 m. What have been the Bentham measurements on these days?
R9: The difference between the Beach site at St-Leu and the PDN location is the height. This is mentioned in the revised manuscript (see L239).
The Bentham instrument was under maintenance during the PDN hike and the St-Leu Beach campaign, hence the data were not available for comparison, unfortunately.
The maximum of UVI recordings made during the experiment at PDF and LEU were 21.1 and 14.5, while Météo-France UVI forecast was 18 and 16, respectively. This difference between the two sites is mainly due to the altitude.
C10: Possible additions:
Why not making hand-held measurements near the Bentham? This will give you clear estimate of variability due to hand-held. Be honest with uncertainties. Well-calibrated SolarLight Model 500 is within 10%, 15% when used operationally. UV Index of 22 will not become lower, because uncertainty goes in both directions, but more credible.
Why not using additionally UV-Index-hours? Especially a second scale/column to Fig.3 and Tab.2? Could give a clearer connection to the UV-Index (65 SED * 10/9 = 72 UVIh)
Saxebøl, UVIh—a proposal for a practical unit for biological effective dose for ultraviolet radiation exposure, Radiat. Prot. Dosim., 2000, 88, 261
Ratio of personal UV expose to ambient UV?
(School Children and outdoor workers (P.6, L.209) are maybe not the target groups for a 3000m hike)
What about sun protection? SPF for hats?
Diffey and Cheesman, BJD 127 (1) 1992 10-12
R10: Thank you for these excellent suggestions and additions. They are welcomed suggestions that we plan to adapt in our future work.

Reviewer 2 Report
The paper 'Solar UV radiation in the tropics: human exposure at Reunion Island (21°S, 55°E) during summer outdoor activities' by Cadet et al. is well organised and has a clear construction. Methodology seems to be adequate to the considered problem and is mostly clearly described. Results are provided in the form of understandable tables and graphs. The work aims to show the threat related to very high UVI at Reunion Island, which is the common destination for tourists. Commonly, tourists, especially from the regions with low or moderate UVI, do not realize that sun can pose such a threat. Authors conducted measurement campaign in the common destinations in the Island. The results show, that UVI can reach even above 22, with the cumulative dose above 60 SED.
General comments:
The paper is well written, with clear construction. The abstract summarize the work adequately, describes the overall work and contains results. The problem raised by the researches is of a great importance to human health. Authors claim, that if the information abour UVI would be in the forecasts, that the public awarness would be raised. I disagree, because UVI is avaiable in almost every forecast (especially smartphone application) but people rarely know what it means. On the other hand, I liked the idea of placing signboards about UVI danger in the area of extremely high UVI, especially in the mountains. More references about exposure ratio is needed in the article. For example:
Weihs, A. Schmalwieser, C. Reinisch, E. Meraner, S. Walisch, M. Harald,Measurements of personal UV exposure on different parts of the body during var-ious activities, Photochem. Photobiol. 89 (2013) 1004–1007,https://doi.org/10.1111/php.12085.
Downs, A. Parisi, Mean exposure fractions of human body solar UV exposurepatterns for application in different ambient climates, Photochem. Photobiol. 88(2012) 223–226,https://doi.org/10.1111/j.1751-1097.2011.01025.x.
Detailed comments:
Line 81 Page 2: "However, UVI is not well known by the population and even by the dermatologists" - do authors mean overall population and dermatologists or from the Island?
Line 95 Page 3: authors could add some references about scientific use of Solarmeter 6.5 by other researchers. The calibration of Solarmeter 6.5 with Bentham spectroradiometer in the SDN location would be a valueable asset.
Line 220 Page 6: TUV reference is needed here
Also, authors could add a figure or table with the maximum time of exposure without sunburn for the highest UVI (~22) and different phototypes to underline how dangerous such UVI is.
Author Response
1 October 2020
Open Review
Dear Reviewer,
First of all, I would like to thank you for your work on our paper. Our response is structured based on your review. We identified two comments, referred hereafter as C1 to C5 and shown in black. Our responses are indeed labelled from R1 to R5 and printed in blue.
To make the reading of the revised manuscript easier, all edits are highlighted in yellow.
With kind regards,
Jean-Maurice CADET
The paper 'Solar UV radiation in the tropics: human exposure at Reunion Island (21°S, 55°E) during summer outdoor activities' by Cadet et al. is well organised and has a clear construction. Methodology seems to be adequate to the considered problem and is mostly clearly described. Results are provided in the form of understandable tables and graphs. The work aims to show the threat related to very high UVI at Reunion Island, which is the common destination for tourists. Commonly, tourists, especially from the regions with low or moderate UVI, do not realize that sun can pose such a threat. Authors conducted measurement campaign in the common destinations in the Island. The results show, that UVI can reach even above 22, with the cumulative dose above 60 SED.
General comments:
C1: The paper is well written, with clear construction. The abstract summarize the work adequately, describes the overall work and contains results. The problem raised by the researches is of a great importance to human health. Authors claim, that if the information abour UVI would be in the forecasts, that the public awarness would be raised. I disagree, because UVI is avaiable in almost every forecast (especially smartphone application) but people rarely know what it means. On the other hand, I liked the idea of placing signboards about UVI danger in the area of extremely high UVI, especially in the mountains. More references about exposure ratio is needed in the article. For example:
Weihs, A. Schmalwieser, C. Reinisch, E. Meraner, S. Walisch, M. Harald, Measurements of personal UV exposure on different parts of the body during various activities, Photochem Photobiol. 89 (2013) 1004–1007,https://doi.org/10.1111/php.12085.
Downs, A. Parisi, Mean exposure fractions of human body solar UV exposure patterns for application in different ambient climates, Photochem. Photobiol. 88(2012) 223–226,https://doi.org/10.1111/j.1751-1097.2011.01025.x.
R1: We thank the reviewer for these suggested references. They are now cited (see L67).
Detailed comments:
C2: Line 81 Page 2: "However, UVI is not well known by the population and even by the dermatologists" - do authors mean overall population and dermatologists or from the Island?
R2: This statement refers to population from the Island. The sentence has been reworded (see L88).
However, UVI is not well known by the local population and with low awareness of the extreme UVI dangers by even the local dermatologists.
C3: Line 95 Page 3: authors could add some references about scientific use of Solarmeter 6.5 by other researchers. The calibration of Solarmeter 6.5 with Bentham spectroradiometer in the SDN location would be a valuable asset.
R3: Before the start of the field UVI campaigns, our Solarimeter instrument went through a side-by-side comparison test with the Bentham spectroradiometer in the SDN location. We did not report on this in the first version of the manuscript. This is now added in the revised version, and the calibration of the Solarmeter with the Bentham appeared to overestimate the UVI values by 12%, which is coherent with the 10% and 5% accuracies given by the NIST (National Institute of Standards and Technology) for the Solarmeter and of the Bentham instruments, respectively. (see L111-114).
C4: Line 220 Page 6: TUV reference is needed here
R4: A TUV reference is added (L232).
These are pre-computed tables generated with the TUV model for multiple values of ozone profile, altitude, solar zenith angle, surface albedo and aerosols [39].
[39] Madronich, S. UV Radiation in the Natural and Perturbed Atmosphere; Lewis Publisher: Boca Raton, FL, USA, 1993.
C5: Also, authors could add a figure or table with the maximum time of exposure without sunburn for the highest UVI (~22) and different phototypes to underline how dangerous such UVI is.
R5: Thank you for the recommendation. However, since the maximum exposure time depends on skin phototype, it will be dedicated in a specific separate study planned for the future.

Reviewer 3 Report
Dear authors,
at first I like to thank you for your contribution to the field of UVR exposure. I like the idea of highlighting spots of high exposure like you did in your manuscript.
But, nevertheless, I have some major points to raise:
- In the introduction, I am missing a short passage on NMSC. This is the most frequent skin cancer induced by UVR and numbers vastly exceed those of MM.
- The numbers you derive are more immission values rather than exposure values. You should spend a passage discussing that in more detail. Exposure is what you receive, UVI is what is around (kind of UV level). Please discuss some points that are linked to the exposure of a distinct body part.
- In addition to the point I raised before, you should discuss that protective clothing easily reduces the exposure.
Some minor points:
- From a technical point of view, UVI are integers, so I suggest to re-read the manuscript for this reason.
- In formulas and text elements, I find the position of the multiplication symbol somewhat misplaced (down at the line instead of usual height).
I am convinced that the manuscript will be an additional mosaic in the big UV picture. Thus it should be accepted for publication, but highly recommend the modifications I suggested.
Author Response
1 October 2020
Open Review
Dear Reviewer,
First of all, I would like to thank you for your work on our paper. Our response is structured based on your review. We identified two comments, referred hereafter as C1 to C5 and shown in black. Our responses are indeed labelled from R1 to R5 and printed in blue.
To make the reading of the revised manuscript easier, all edits are highlighted in yellow.
With kind regards,
Jean-Maurice CADET
At first I like to thank you for your contribution to the field of UVR exposure. I like the idea of highlighting spots of high exposure like you did in your manuscript.
But, nevertheless, I have some major points to raise:
C1: In the introduction, I am missing a short passage on NMSC. This is the most frequent skin cancer induced by UVR and numbers vastly exceed those of MM.
R1: NMSC and MM are mentioned in the revised version, see L63
The characteristics of UVR increase the risk non-melanoma and melanoma skin cancer, eye disease, such as cataracts or immunodeficiency when people are over exposed to UVR [10,11].
C2: The numbers you derive are more immission values rather than exposure values. You should spend a passage discussing that in more detail. Exposure is what you receive, UVI is what is around (kind of UV level). Please discuss some points that are linked to the exposure of a distinct body part.
R2: We thank the reviewer for the comment. By immission, we assume the reviewer means irradiation. In section 2.1.1, we explain that the instrument measures irradiation and then converts that into a UVI exposure metric. This appears in lines L106-107:
The instrument records erythemal-weighted UVR and the UVI - as a proxy for exposure that is more readily understood by the public - is calculated following the standard formula (Equation 1) [27]. Erythemal-weighted UVR is obtained by integrated solar irradiance with erythemal action spectrum.
C3: In addition to the point I raised before, you should discuss that protective clothing easily reduces the exposure.
R3: This was incorporated at L66-67 and L309.
The risk related to UVR largely depends on skin phototype which characterizes sunburn susceptibility (Table 1 shows the Fitzpatrick Skin Phototype classification [12]) and also on protective measures such as adequate clothing, sunscreen, hats or sunglasses [13,14].
Along with more awareness campaigns and more efficient measuring devices placed around the island, a general understanding of UVR will improve, thereby reducing the risks linked to excess exposure resulting from insufficient UVR protection (adequate clothing, sunglasses, hats, sunscreen).
Some minor points:
C4: From a technical point of view, UVI are integers, so I suggest to re-read the manuscript for this reason.
R4: As the reviewer points out, for communication and awareness purposes with the general public, the values of the UVI are integers. However, many UV sensors allow measurements with decimal numbers, depending on accuracy. In our study we used UVI values as displayed by our Solarmeter instrument. This avoids increasing the uncertainty of the measurements.
C5: In formulas and text elements, I find the position of the multiplication symbol somewhat misplaced (down at the line instead of usual height).
R5: All of the formulas have been checked for better positioning.
I am convinced that the manuscript will be an additional mosaic in the big UV picture. Thus it should be accepted for publication, but highly recommend the modifications I suggested.
Reviewer 4 Report
The authors seek to establish the dangerously high levels of UV exposure attained in a popular vacation spot of Reunion Island during commonly performed activities. While this is admirable and the study provides more than adequate justification for the proposed educational awareness campaign, I do have several concerns and suggestions to hopefully improve the manuscript as described below. Many of my comments relate to English language and style.
- Line 23: The UVI range on Reunion Island is stated as high as 8 in winter and 16 in summer. Later in the manuscript (line 79) a higher maximum UVI of 20 is mentioned as previously documented by Cadet et al. Please clarify.
- Line 54: Where it is stated “Vitamin D has also a huge impact on…..” I suggest replacement of the word ‘huge’ with ‘substantial’ or other term more commonly used in scientific writing.
- Line 59: The statement “The risk related to UVR depends on skin phototype…” should be revised to add it ‘largely’ depends on skin phototype as this statement ignores the fact that use of protective measures also influence risk. Consider adding a statement to describe the potential influence of protective measures on UV exposure risks such as UV protective clothing, sunscreen, hats, and sunglasses.
- Lines 65-66: This statement is quite obvious, yet "addictive" is not commonly used to refer to sunbed use. Perhaps combine this concept with the preceding sentence. Something like "Frequent use of sunbeds obviously increases UVR exposure and this behavior has been linked to acceleration of skin aging."
- Line 70: While the term ‘endemism’ is correct, it is not a commonly used term. Consider revising.
- Line 80: Consider rephrasing - 'even by the dermatologists' to 'with low awareness of the extreme UVI dangers by even the local dermatologists.'
- Line 90 (Table 1): The Fitzpatrick skin phototype classification table requires revision. Specifically, Types III, IV, and V skin descriptions are different than usual classification. Revise to more accurately describe: Type III should read ‘light brown skin’; Type IV should read ‘moderate brown skin’; Type V ‘dark brown skin’; Type VI ‘dark brown to black skin’. Alternatively, provide reference to support current descriptions.
- Line 133: Change ‘volcano’ to ‘volcanoes’.
- Line 155: Rather than ‘will be computed’ this should be written as past tense ‘were computed’.
- Lines 164 – 178: The term ‘relief’ is used several times within this paragraph. From the context, I gather that ‘shade’ is meant; it is suggested use of the more common term ‘shade’ in English usage.
- Lines 254-261 (Figure 3): It is stated UVI index histograms use 'standard UVI color scale' yet most UVI scales end at violet to indicate UVI of 11 or greater. It is clear from the graphs that light blue (or light violet?) is prominently used for UVI exceeding 11. Please elaborate on the use of additional color increments and/or provide a reference to support this expanded color scale.
- Lines 276-277: Consider adding a phrase such as ‘during each individual popular activity’ to the end of the statement “The UVR assessment performed revealed that total UVR exposure can reach 65 SED.”
- Line 283: Consider changing ‘making them, unfortunately, less accessible or known’ to 'unfortunately making them less accessible or known.'
- Line 290: Change ‘sun bath’ to ‘sunbathing’.
- Lines 298-299: Recommend rephrasing of the concluding statement to rather than minimize, emphasize, as recognizing generalizability to other similar climates is quite important rather than ‘needless to say.’
Author Response
1 October 2020
Open Review
Dear Reviewer,
First of all, I would like to thank you for your work on our paper. Our response is structured based on your review. We identified two comments, referred hereafter as C1 to C15 and shown in black. Our responses are indeed labelled from R1 to R15 and printed in blue.
To make the reading of the revised manuscript easier, all edits are highlighted in yellow.
With kind regards,
Jean-Maurice CADET
The authors seek to establish the dangerously high levels of UV exposure attained in a popular vacation spot of Reunion Island during commonly performed activities. While this is admirable and the study provides more than adequate justification for the proposed educational awareness campaign, I do have several concerns and suggestions to hopefully improve the manuscript as described below. Many of my comments relate to English language and style.
C1: Line 23: The UVI range on Reunion Island is stated as high as 8 in winter and 16 in summer. Later in the manuscript (line 79) a higher maximum UVI of 20 is mentioned as previously documented by Cadet et al. Please clarify.
R1: This is mean climatological UVI: 8 in winter and 16 in summer. This information has been added in L28.
Reunion is known to have high levels of solar ultraviolet radiation (UVR) with an ultraviolet index (UVI) which can reach 8 in winter and 16 in summer (mean climatological conditions).
C2: Line 54: Where it is stated “Vitamin D has also a huge impact on…..” I suggest replacement of the word ‘huge’ with ‘substantial’ or other term more commonly used in scientific writing.
R2: Word has been changed in L59
Vitamin D has also a substantial impact on brain chemistry, for example in brain serotonin levels which fight anxiety and depression [7]. In medicine, UVR has been used in phototherapy for decades [8,9].
C3: Line 59: The statement “The risk related to UVR depends on skin phototype…” should be revised to add it ‘largely’ depends on skin phototype as this statement ignores the fact that use of protective measures also influence risk. Consider adding a statement to describe the potential influence of protective measures on UV exposure risks such as UV protective clothing, sunscreen, hats, and sunglasses.
R3: These sentences have been rephrased in lines L64-67.
The risk related to UVR largely depends on skin phototype which characterizes sunburn susceptibility (Table 1 shows the Fitzpatrick Skin Phototype classification [12]) and also on protective measures such as adequate clothing, sunscreen, hats or sunglasses [13,14].
C4: Lines 65-66: This statement is quite obvious, yet "addictive" is not commonly used to refer to sunbed use. Perhaps combine this concept with the preceding sentence. Something like "Frequent use of sunbeds obviously increases UVR exposure and this behavior has been linked to acceleration of skin aging."
R4: The sentence has been rephrased in lines L72-73.
Moreover, frequent use of sunbeds may become addictive, and subsequently increases UVR exposure where this behavior has been linked to acceleration of skin aging [19].
C5: Line 70: While the term ‘endemism’ is correct, it is not a commonly used term. Consider revising.
R5: We have amended this sentence:
Listed as a world UNESCO heritage site since 2010, the wide biodiversity of endemic plants and animals generates a great interest for the destination [20].
C6: Line 80: Consider rephrasing - 'even by the dermatologists' to 'with low awareness of the extreme UVI dangers by even the local dermatologists.'
R6: We have rephrased this sentence rephrased L87-88:
However, UVI is not well known by the local population and there is even a low awareness of the extreme UVI dangers by the local dermatologists [25].
C7: Line 90 (Table 1): The Fitzpatrick skin phototype classification table requires revision. Specifically, Types III, IV, and V skin descriptions are different than usual classification. Revise to more accurately describe: Type III should read ‘light brown skin’; Type IV should read ‘moderate brown skin’; Type V ‘dark brown skin’; Type VI ‘dark brown to black skin’. Alternatively, provide reference to support current descriptions.
R7: We have revised Table 1 according to reference [12].
Table 1: Fitzpatrick Skin phototype classification [12]
Phototype |
Characteristics |
History of sunburn |
Minimal dose to elicit sunburn (SED) |
I |
Ivory white skin, light eyes |
Burns easily |
2-3 |
II |
White skin, hazel/brown eyes |
Burns easily |
2.5-3 |
III |
White skin, brown eyes |
Burns moderately |
3-5 |
IV |
Light brown skin, dark eyes |
Burns minimally |
4.5-6 |
V |
Brown skin, dark eyes |
Rarely burns |
6-20 |
VI |
Dark brown skin, dark eyes |
Never burns |
6-20 |
C8: Line 133: Change ‘volcano’ to ‘volcanoes’.
R8: We have changed ‘volcano’ to ‘volcanoes’, see L136 and L144.
C9: Line 155: Rather than ‘will be computed’ this should be written as past tense ‘were computed’.
R9: This was corrected to “were calculated” in L162 and were computed in Line 169.
C10: Lines 164 – 178: The term ‘relief’ is used several times within this paragraph. From the context, I gather that ‘shade’ is meant; it is suggested use of the more common term ‘shade’ in English usage.
R10: We replaced the term “relief” by “shade” within the paragraph as suggested in lines L180, L188 and L190.
C11: Lines 254-261 (Figure 3): It is stated UVI index histograms use 'standard UVI color scale' yet most UVI scales end at violet to indicate UVI of 11 or greater. It is clear from the graphs that light blue (or light violet?) is prominently used for UVI exceeding 11. Please elaborate on the use of additional color increments and/or provide a reference to support this expanded color scale.
R11: The elaboration of our graphs was based on an adaptation of the WHO colour scale which assigns a colour code to UVI values. We have amended our figure caption to explain this in L268-270.
The colors of the histograms are similar to those used by the WHO, but with some variations within each color-band to show more details.
C12: Lines 276-277: Consider adding a phrase such as ‘during each individual popular activity’ to the end of the statement “The UVR assessment performed revealed that total UVR exposure can reach 65 SED.”
R12: We have amended the sentence as suggested (see L292).
The UVR assessment performed revealed that total UVR exposure can reach 65 SED during each individual popular activity.
C13: Line 283: Consider changing ‘making them, unfortunately, less accessible or known’ to 'unfortunately making them less accessible or known.'
R13: We have amended the sentence as suggested (see L299).
Anecdotally, most people are usually not informed well of the dangers of long-term exposure to UVR especially in a tropical environment. Météo-France provides UVI forecasts [5], but this information is only available on Météo-France website, unfortunately making them less accessible or known
C14: Line 290: Change ‘sun bath’ to ‘sunbathing’.
R14: This has been corrected (see L306).
C15: Lines 298-299: Recommend rephrasing of the concluding statement to rather than minimize, emphasize, as recognizing generalizability to other similar climates is quite important rather than ‘needless to say.’
R15: The concluding statement was rephrased as suggested (see L314):
It is important to emphasize that Reunion observations and the above recommendations may be relevant to many other sites in tropical countries and territories.

Round 2
Reviewer 3 Report
Dear authors,
thank you for considering my comments. Although some improvements to the manuscript have been made, some of my points have not/not fully been adressed:
- in the introduction I still miss some additional information on NMSC and UVR, especially on how UVR causes NMSC
- The multiplication sign is still misplaced, e.g. in formula 1, l116, l163, l165
- My comment C2 has not been adressed properly: please explain the difference between measurements of the UV radiation on a flat surface to obtain the UVI and the real exposure of the skin, which is also mainly by movement etc.. This is what I meant by the difference between immission and exposure
Sorry in case I missed it!
Author Response
15 October 2020
Open Review
Dear Reviewer,
First of all, I would like to thank you for your work on our paper. Our response is structured based on your review. We identified three comments, referred hereafter as C1, C2 and C3 and shown in black. Our responses are indeed labelled from R1 to R3 and printed in blue.
To make the reading of the revised manuscript easier, all edits are highlighted in yellow for the 1st round revision and in green for the 2nd round.
With kind regards,
Jean-Maurice CADET
Dear authors,
thank you for considering my comments. Although some improvements to the manuscript have been made, some of my points have not/not fully been adressed:
C1: in the introduction I still miss some additional information on NMSC and UVR, especially on how UVR causes NMSC
R1: The introduction was modified to include additional information on NMSC and UVR as suggested (see L62-66 in the revised manuscript).
The characteristics of UVR increase the risk of sunburn, eye disease, such as cataracts, or immunodeficiency when people are over exposed to UVR. The skin carcinogenesis effect of UVR increases the risk of non-melanoma skin cancer (NMSC) by DNA damage and rapid, abnormal increase of keratinocytes [10-12].
C2: The multiplication sign is still misplaced, e.g. in formula 1, l116, l163, l165
R2: The formulas was checked.
C3: My comment C2 has not been adressed properly: please explain the difference between measurements of the UV radiation on a flat surface to obtain the UVI and the real exposure of the skin, which is also mainly by movement etc.. This is what I meant by the difference between immission and exposure. Sorry in case I missed it!
R3: We only measure ambient (immission) close to the operator and no skin exposure on a particular part of the body. This is a limitation of our study. We mentioned it in the manuscript (L157-158).
Ambient erythemal UVR measurements were made using a handheld Solarmeter Model 6.5 UV Index Meter (SN#10414). The latter does not allow to measure skin exposure for any part of the body.
